# Recollection of Physician Information about Risk Factor and Lifestyle Changes in Chronic Coronary Syndrome Patients

**DOI:** 10.3390/ijerph19116416

**Published:** 2022-05-25

**Authors:** Siamala Sinnadurai, Pawel Sowa, Piotr Jankowski, Zbigniew Gasior, Dariusz A. Kosior, Maciej Haberka, Danuta Czarnecka, Andrzej Pajak, Malgorzata Setny, Jacek Jamiolkowski, Emilia Sawicka-Śmiarowska, Karol Kaminski

**Affiliations:** 1Department of Population Medicine and Lifestyle Diseases Prevention, Faculty of Medicine with the Division of Dentistry and Division of Medical Education in English, Medical University of Bialystok, 15-089 Bialystok, Poland; siamala.sinnadurai@umb.edu.pl (S.S.); pawel.sowa@umb.edu.pl (P.S.); jacek909@wp.pl (J.J.); emiliasawickak@gmail.com (E.S.-Ś.); 2Department of Internal Medicine and Geriatric Cardiology, Center of Postgraduate Medical Education, 00-416 Warszawa, Poland; piotrjankowski@interia.pl; 3Department of Epidemiology and Health Promotion, School of Public Health, Center of Postgraduate Medical Education, 01-813 Warszawa, Poland; 4Department of Cardiology, School of Health Sciences, Medical University Silesia, 40-055 Katowice, Poland; zgasior@ptkardio.pl (Z.G.); mhaberka@op.pl (M.H.); 5Department of Cardiology and Hypertension with Electrophysiology Lab, Central Research Hospital of the Ministry of the Interior and Administration, 02-507 Warsaw, Poland; dariusz.kosior@cskmswia.pl (D.A.K.); malgoskub@gmail.com (M.S.); 6Mossakowski Medical Research Institute, Polish Academy of Sciences, 02-106 Warsaw, Poland; 7Department of Cardiology, Interventional Electrocardiology and Hypertension, Institute of Cardiology, Jagiellonian University Medical College, 31-008 Kracow, Poland; danuta.czarnecka@uj.edu.pl; 8Department of Epidemiology and Population Studies, Institute of Public Health Faculty of Health Sciences, Jagiellonian University Medical College, 31-066 Krakow, Poland; andrzej.pajak@uj.edu.pl; 9Department of Cardiology, Medical University of Bialystok, 15-276 Bialystok, Poland

**Keywords:** chronic coronary syndrome, communication, risk factor information

## Abstract

A patient’s compliance to a physician’s lifestyle information is essential in chronic coronary syndrome (CCS) patients. We assessed potential characteristics associated with a patient’s recollection of physician information and lifestyle changes. This study recruited and interviewed patients (aged ≤ 80 years) 6–18 months after hospitalization due to acute coronary syndrome or elective myocardial revascularization. A physician’s information on risk factors was recognized if patients recollected the assessment of their diet, weight management, blood pressure control, cholesterol level, diabetes, and other lifestyle factors by the doctor. Of a total of 946 chronic coronary syndrome patients, 52.9% (501) of them declared the recollection of providing information on more than 80% of the risk factors. A good recollection of risk factor information was associated with the following: a patient’s age (OR per year: 0.97; 95% CI: 0.95 to 0.99), obesity (OR: 4.41; 95% CI: 3.09–6.30), diabetes (OR: 4.16; 95% CI: 2.96–5.84), diuretic therapy (OR: 1.41; 95% CI: 1.03–1.91), calcium channel blockers (OR: 1.47; 95% CI: 1.04–2.09), and ACEI/sartan (OR: 0.65; 95% CI: 0.45–0.94) at hospitalization discharge. In terms of goal attainment, better adherence to antihypertensive drugs (OR: 1.80; 95% CI: 1.07–3.03) was observed in the patients with a good compared to a poor recollection of risk factor information. The recollection of physician risk factor information was significantly associated with more comorbidities. Strategies to tailor the conveying of information to a patient’s perception are needed for optimal patient–doctor communication.

## 1. Introduction

A patient’s unhealthy lifestyle is known to be the most important modifiable risk factor for the majority of deaths from coronary heart disease (CHD) [1,2]. To reduce the subsequent risk of CHD, healthcare professionals often seek an intervention focused on lifestyle modifications, especially in terms of diet, physical exercise, smoking, and weight [3]. With this being the case, an accumulating amount of evidence supports the notion that lifestyle interventions significantly reduce the risk of CHD events [4,5,6,7,8]. Meanwhile, in real life, these interventions are moderate but potentially yield relevant effects in these patients. Lifestyle interventions are most effective when a patient is cooperative and fully involved in the lifestyle modification process, which includes them stopping smoking, practicing a healthy diet, regularly exercising, and monitoring signs of high blood pressure, as well as cholesterol. Yet, adherence to lifestyle advice in clinical practice is not optimal [9].

In Europe, the present prevention approaches of coronary heart disease include advice on changing lifestyles and risk factor management, according to the European Society of Cardiology [5]. Repeatedly, a cardiac rehabilitation and education program has been addressed as a core part of secondary prevention control [10,11]; however, at the time of our study, less than half of the coronary patients had been offered participation in such a program [12]. The recent EUROASPIRE V revealed that the majority of coronary patients have unhealthy lifestyles with low risk factor control (e.g., blood pressure, low-density lipoprotein cholesterol, and glucose targets) [13]. This indicated that there is an urgent need for the implementation of structured, managed care for patients with coronary heart disease, which should be scrutinized by continuing to monitor lifestyle behavior changes to keep cardiovascular risk factors under control.

Patient–doctor interactions are complex, and communicating information about a disease to a patient is challenging [14]. Nevertheless, a significant reduction in an individual’s CHD risk requires an appropriate assessment of the risk and effective communication of said risk to anticipate risk factor treatment. Encouraging or motivating patients to change their lifestyle habits requires skills in behavioral science and ample time for physicians to explain the importance of doing so. For instance, this could involve the incorporation of “soft skills” on how to communicate effectively with patients, as well as how to share clinical evidence and explore ways of educating patients to take responsibility and engage in their own care. Additionally, the risk of disease on its own is not effective and needs to be coupled with other intervention elements to promote healthy behavior [15]. Thus, effective communication skills to convey risk in a comprehensible way may provide an important step in obtaining favorable changes in patients’ lifestyle habits to combat the subsequent adverse event of the disease [16]. Doctors must also take into consideration a patient’s perception of one’s health in addition to the predisposition for downplaying possible health threats in some demographic groups (e.g., apparently healthy middle-aged men).

To our knowledge, there are only a few studies that focus on patient–doctor communication in promoting healthy lifestyle behaviors in Europe [17]. Moreover, in clinical practice, not all physicians are well-trained to provide crucial advice to specific groups of patients. Therefore, we planned a study on the efficacy of patient–doctor communication by examining the prevalence of and potential determinants associated with the recollection of lifestyle-associated risk factor information by patients with coronary heart disease. In addition, we sought to address their impact on risk factor goal achievement and lifestyle behavioral changes in secondary prevention.

## 2. Materials and Methods

### 2.1. Study Design and Patient Population

We used the POLASPIRE (Polish Action on Secondary and Primary Prevention by Intervention to Reduce Events) database, a Polish survey that contributed to the EUROASPIRE V study, which aimed to recruit coronary heart disease patients from 2016 to 2017 in Poland. This is a cross-sectional study, and it involved 14 hospitals in 4 regions: Cracow, Katowice, Warsaw, and Bialystok. The study screened patients aged >18 years and <80 years who had been hospitalized ≥6 months to <2 years prior due to (i) an elective or emergency coronary artery bypass graft (CABG), (ii) an elective or emergency percutaneous coronary intervention (PCI), (iii) acute myocardial infarction (International Classification of Diseases, Tenth Revision, codes I21 and I22), or (iv) unstable angina in the participating hospitals for eligibility. These patients were invited to participate in a face-to face interview as part of the study. The selection procedure was performed by centrally trained experts. The details of the study methods have been published elsewhere [9,18,19].

### 2.2. Study Variables

Study variables were collected at two time points: hospitalization (from discharge letters) and at the time of the interview (using EUROASPIRE V questionnaires). These included the presence of risk factors, anthropometric measurements (body weight, height, and waist circumference), blood pressure values, the results of biochemical tests, such as glucose, HBA1c, creatinine, and plasma lipids, the history of CVD-related procedures performed, and the treatment prescribed. Educational level was defined as completion of last education level. Body mass index was calculated as weight divided by height squared, kg/m^2^. A patient was defined as obese if their body mass index (BMI) was ≥30 kg/m^2^. Increased blood pressure was defined as blood pressure ≥ 140/90 mm Hg. Patients with cholesterol levels ≥ 1.8 mmol/L were noted as having high blood cholesterol, and a controlled blood glucose level (HbA1c) was accepted as that which was lower than 7%, as per guidelines. A current smoker was defined as a patient that reported themselves to be a smoker in the month before the index event, as well as one whose breath carbon monoxide exceeded 10 ppm at the time of the interview. Medication was obtained from discharge letters and was self-reported by patients at the interview. Self-reported information on psychosocial function—depression and anxiety—was assessed using the Hospital Anxiety and Depression Scale (HADS), where a HADS score lower than 8 points was considered as normal. Quality of life was determined by using the Heart-Related Quality of Life (HRQL) questionnaire, which consists of two domains divided by physical (10 items) and emotional (4 items); by calculating the total scores of each, we defined the highest HRQL. In the Visual Analogue Scale (VAS), the score ranged from 0 to 100, and the highest value described a patient’s best possible health. 

### 2.3. Strategy in Defining the Exposure Group 

The recollection of a physician’s information on lifestyle and CVD risk factor management, based on the European Guidelines on CVD Prevention in Clinical Practice [16], was measured via the Polish version of the EUROASPIRE survey questionnaire [12]. A patient’s recollection of a physician’s clinical risk factor information was categorized based on the amount of information on the management of particular risk factors that the patient recalled from the index hospitalization or cardiac rehabilitation program. These questions were as follows: (1) have you been told SINCE the index event by a health care professional that your diet is unhealthy; (2) have you ever been told by health care professional that you are overweight; (3) have you ever been told by doctor (or other health professional) that you have high blood pressure; (4) have you ever been told by doctor (or other health professional) that you have high blood cholesterol; (5) have you have received lifestyle advices from other professional group; and (6) have you ever been told by a doctor or other health professional that you have diabetes. We calculated the rate of recollection of risk factor and lifestyle items based on a patient’s information about risk factor and lifestyle items divided by the potential information or lifestyle items the patient was eligible for. Good recollection of risk factor information (GRRFI) was categorized if a patient recollected risk factors and lifestyle items at a higher rate than the median of all the survey responses, ≥80%, and poor recollection of risk factor information (PRRFI) items was when a patient did so at a lower rate than the median, <80%. 

### 2.4. Risk Factor Goal Achievement

The primary assessment was the determinants associated with patient recollection of a physician’s risk factor information. The secondary assessment was the measures of self-reported risk factor goal achievement as defined in the 2016 ESC guidelines [6], namely for weight (BMI), blood pressure (SBP and DBP), lipid profile (LDL, HDL, and triglycerides), glucose level (HbA1c), medication adherence, and motivation to change their behavior (e.g., in smoking habits, intake of a healthy diet, weight management, level of physical activity, blood pressure, and cholesterol control strategy). Additionally, we assessed changes in the clinical parameters (e.g., BMI, LDL, HDL, triglycerides, SBP, DBP, and medication intake) between the baseline (at the time of hospitalization) and after 6–18 months (an interview).

### 2.5. Ethical Statement

All patients provided written informed consent to take part in the study. The study was approved by the local ethics committees in each regional center. 

### 2.6. Data Analysis

Categorical variables were reported in proportions (%), and continuous variables in means and standard deviations. Data on demographics, clinical characteristics, and medication administration were compared between GRRFI and PRRFI using the chi-square (categorical) and Mann–Whitney U tests (continuous). All variables with a *p*-value of <0.25 in the univariable analyses were included in a backward multivariable logistic regression model to identify factors that were independently associated with a good recollection of a physician’s risk factor information. A linear regression model was used to analyze the differences between clinical parameters at the baseline and the interview, as well as the recollection of a physician’s risk factor information status. A two-tailed *p*-value of less than 0.05 and a 95% CI for an odds ratio that did not include 1 were considered statistically significant. All analyses were performed using International Business Machines SPSS Statistics software, version 28 (IBM, Armonk, New York, NY, USA). 

## 3. Results

### 3.1. Characteristics of the Study Population 

The final analysis included 946 chronic coronary syndrome patients after excluding 72 who did not complete the risk factor questionnaire during the face-to-face interview. More than one-third of the hospitalizations in the study cohort were due to a percutaneous coronary intervention (PCI) (37.5%); less than sixty percent had myocardial infarction, namely unstable angina (21.6%), non-ST-segment elevation myocardial infarction (21.1%), and ST-segment elevation myocardial infarction (15.6%), and the least common cause for hospitalization was coronary artery bypass grafting (4.0%). In general, the proportion of recollection of a physician’s risk factor information reached nearly fifty percent or more in all of the items included in study: high blood pressure (80.1%), high blood cholesterol (73.0%), increased weight (47.9%), diet (48.2%), and received advice from other professional group (81.4%) (*not shown in table*). More than three-fourths of the coronary heart disease patients had risk factors, such as hypertension and hyperlipidemia, at the time of hospitalization (Table 1). 

### 3.2. Patient Characteristics Compared with the Recollection of Risk Factor Information Status (Good Recollection Versus Poor Recollection)

The majority of CCS patients were elderly, with a median age at diagnosis of 65 years (with an interquartile range of 60 to 71), male (71.1%), underwent percutaneous coronary intervention (PCI) (37.5%), were examined in a teaching hospital (82.2%), and had obtained middle-level education (67.6%) (Table 1). Slightly more than half of the patients were identified as having a good recollection of risk factor information (*n* = 501). In addition, good recollection of risk factor information (GRRFI) was significantly associated with self-reported obesity (*p* < 0.001), hypertension (*p* < 0.001), hyperlipidemia (*p* < 0.001), diabetes (*p* < 0.001), and the prescription of calcium channel blockers or diuretics (*p* < 0.001) compared with those with a poor recollection of risk factor information (PRRFI). Education status and smoking habit were not related to the amount of information recalled. A multivariable model was created by including covariates (a patient’s demographics, clinical characteristics, and medication intake) as the independent variables and the recollection of risk factor information (RRFI) status as the dependent variable (PRRFI = 0 and GRRFI = 1). Additionally, by incorporating backward logistics model selection, patients that presented with cardiovascular disease risk factors, such as being obese (OR: 4.41; 95% CI: 3.09–6.30) or having diabetes (OR: 4.16; 95% CI: 2.96–5.84), at the time of hospitalization were independently associated with a higher chance of having a GRRFI compared to their PRRFI counterparts (Table 2). Compared with PRRFI, GRRFI was significantly associated with favorable hypertensive medication administration, such as calcium channel blockers (OR: 1.47; 95% CI: 1.04–2.09) and diuretics (OR: 1.41; 95% CI: 1.03–1.91). (Table 2). The recollection of risk factor information decreased with age (*p*_trend_ = 0.002), and patients aged ≥ 65 years were fifty percent less likely to recollect risk factor information compared to younger patients.

In an adjusted multivariable model, created by including RRFI status (PRRFI = 0 and GRRFI = 1) and covariates as the independent variables and risk factor goals as the dependent variables, no statistically significant improvement in secondary prevention goal achievement was observed in patients with a GRRFI compared to those with a PRRFI (Table 2). Compared with PRRFI, GRRFI possesses favorable medication adherence, with nearly a two-fold increase in completing > 75 % of prescribed antihypertensive drugs (OR: 1.80; 95% CI: 1.07–3.03) when the interview was observed. However, quality of life was not improved upon recollecting information on many clinical risk factors and adherence to medication in GRRFI compared with PRRFI. A significant negative association, particularly in heart-related quality of life, global and emotional (β: −0.49; 95% CI: −0.94 to −0.05), was observed in patients with a GRRFI compared with those with a PRRFI (Table 2).

### 3.3. Cardiovascular Parameter Changes between RRFI Status and Different Time Points on Risk Factors and Medication Intake

There was no difference in clinical risk factors between patients with a GRRFI and a PRRFI in an unadjusted multivariable model (Table 3). Nevertheless, a multivariable adjusted model revealed a significant association between the BMI (β: 0.41; 95% CI: 0.07 to 0.75) and GRRFI, which increases when BMI does, meaning that there is a significant interaction between body weight and GRRFI (Table 3).

Medication intake, especially for beta-blockers, was significantly associated with GRRFI compared with PRRFI, by two-fold; however, the association changed in the multivariable adjusted model (Table 4).

### 3.4. Recollection of Risk Factor Information and Lifestyle Changes

Overall, GRRFI was associated with lifestyle improvements compared with PRRFI in terms of reduction in salt intake (OR: 1.39; 95% CI: 1.02–1.90), reduction in fat intake (OR: 1.90; 95% CI: 1.31–2.76), reduction in sugar intake (OR: 1.63; 95% CI: 1.19–2.24), increase in fruit and vegetable intake (OR: 1.55; 95% CI: 1.11–2.15), increase in fish intake (OR: 1.37; 95% CI: 1.02–1.84), excess alcohol intake (OR: 1.76; 95% CI: 1.34–2.29), actively trying to lose weight (OR: 1.40; 95% CI: 1.02–1.91), actively trying to keep from gaining weight (OR: 1.58; 95% CI: 1.16–2.15), following dietary recommendations (OR: 1.39; 95% CI: 1.03–1.87), and following a special diet to lower blood pressure (OR: 5.87; 95% CI: 3.27–10.54) (Figure 1 and Figure 2).

In a subgroup analysis of the differences between age, gender, and GRRFI, there was no significant interaction observed between younger and older patients in terms of any lifestyle behavior changes in a univariable crude model (Figure 3) or a multivariable adjusted model (Figure 4).

On the other hand, we observed a significant interaction between diet characteristics, especially a reduction in fat intake (*p* = 0.019), and gender in a univariable crude model (Figure 5). In a multivariable adjusted model, this interaction remained unchanged (*p* = 0.018) (Figure 6).

## 4. Discussion

The recollection of information on risk factors in the secondary prevention patients can still be improved. The recollection of lifestyle advice was the lowest in the oldest group and the highest in the middle-aged patients. Being obese or having diabetes was independently associated with a good recollection of a physician’s information on risk factors. Patients with a GRRFI had worse risk factor control compared to those with a PRRFI. Despite better treatment adherence to antihypertensive drugs being observed in patients with a GRRFI, there was no significant improvement in the quality of life in heart QoL global, physical, and emotional compared with those with a PRRFI, but this may be due to the short observation time. The recollection of lifestyle advice demonstrated no difference in CVD risk factors between hospitalization and the time of the interview. While lifestyle behavior changed significantly with GRRFI between hospitalization and the time of the interview, there was no significant interaction in the age subgroups, but the pattern of fat intake was significantly different between females and males.

The recommendations for coronary heart disease management [16,20] include encouraging physicians and patients to be involved in clinical decision-making on suitable interventions, hence making lifestyle changes more effective. Sometimes, physicians could be less inclined to refer the elderly for an official recommended target intervention (e.g., a cardiac rehabilitation program) [21] due to the reason of frailty and other logistical problems in commuting to a hospital; hence, home-based lifestyle changes would be the best secondary preventive approach. A recent meta-analysis included 22 studies that assessed the association between lifestyle indices and CVD risk factors and revealed that the adoption of several healthy lifestyle behaviors showed a 66% reduction in CVD risks compared to the adoption of at least one or no healthy lifestyle behaviors [22]. This study showed that physicians were more likely to offer lifestyle interventions to older adults, which yielded the most favorable CVD outcomes. However, our results revealed that elderly patients have a lower frequency of a GRRFI. This being the case, a physician’s knowledge and skills are not sufficient as it could be a challenge for them to convey and explain all of the necessary information to elderly patients. On the other hand, the problem could be the way that patients understand and grasp what they are told, as well as the possibility of remembering all the information given to them [23]. Importantly, elderly patients are more prone to be in a state of denial about healthy eating habits or how to stay healthy in general compared with younger patients [24]. Therefore, an action or communication on how to improve lifestyle interventions should be followed by highlighting its necessity to elderly patients to avoid some confusion.

We observed that patients with multiple CVD risk factors were independently associated with a GRRFI. The evidence of a higher frequency of recollecting information in obese and diabetic patients than their PRRFI counterparts adds value, indicating that CHD patients with existing comorbidities overestimate CVD-related adverse outcomes, triggering them to be very cautious and disciplined in regard to their health condition. In contrast, in the study reported by Tiffe et al., only diabetes mellitus was independently associated with receiving appropriate physician lifestyle advice in secondary prevention, whereas patients with diabetes, hyperlipidemia, and hypertension were independently associated with receiving a physician’s lifestyle advice in primary prevention settings [17]. As in the prior study by Tiffe et al. [17] typically measuring the impact of a physician’s lifestyle advice (PLA) using a threshold of less or more than 50%, it remained unknown whether the threshold used was appropriate for secondary prevention settings as the study conducted compared PLA between primary and secondary settings. Importantly, our data suggest that patients in a secondary preventive setting with multiple comorbidities are more likely to recollect much clinical information, suggesting that patients’ satisfaction in patient–doctor interactions provides great awareness of their disease conditions and associated risks. It should be kept in mind that observational studies often suffer from confounding factors or unmeasured factors; therefore, in terms of the statistical aspect, we present results from a step-wise multivariable adjusted model and a backward multivariable adjusted model to provide more intuitive evidence to report factors associated with the recollection of risk factor information.

A further noteworthy finding from our study was that patients with a GRRFI are nearly two-fold more likely to adhere to medication, and antihypertensive drugs in particular had a statistically significant effect on most of their lifestyle changes in the adjusted multivariable model (Figure 2). However, there were no significant age and gender differences in lifestyle changes and risk factor awareness, except for the fact that fewer men reduced their fat intake since being discharged from hospital than women. Generally, women are less likely to take medication than men, especially statin treatments [25,26], due to the possible side effects of drugs [27,28]. Moreover, based on our results, they were also less focused on lifestyle interventions in the area of limiting fat intake. Despite the fact that studies in “real world” populations have demonstrated that adherence to medical advice has a positive impact on CVD outcomes, with subsequently reduced rates of recurrent events [29,30,31], we observed that there were no changes in blood pressure, blood lipids, or body weight, and most of the patients did not achieve their secondary prevention goals. Our findings are in line with a recent report from the European Society of Cardiology, the EUROASPIRE IV survey 12, and even with their latest one, with a larger sample size from 81 regions in 27 European countries, the EUROASPIRE V registry [13], which reported that the majority of coronary patients did not achieve their risk factor goals in terms of blood pressure, low-density lipoprotein, cholesterol, and glucose targets. 

Our report, based on the Polish population together with the results of multiple editions of the EUROASPIRE studies, highly encourages the implementation of new guidelines for CVD prevention in clinical practice in Europe. Specifically, in Poland, a managed-care program was introduced after the POLASPIRE study, and it was appreciated for its initiation. This approach could give a glimpse to cardiology physicians to assess the effect of the program by considering the POLASPIRE study as a control group and those registered in the managed-care program study as an intervention group. Changes in risk factor parameters (e.g., weight, blood pressure and lipid profile, smoking, and exercise) between these groups could provide some insights for improving the secondary preventive strategies. On the other hand, in terms of behavioral changes and also to motivate patients to change their lifestyles, comprehensive risk communication is very important. Taken together with the above discussion, it is important to explore the risk communication topic deliberately to facilitate the implementation of new recommendations and guidelines to improve secondary preventive strategies in chronic coronary syndrome patients.

This study possesses several limitations. First, the data on lifestyle recommendations from physicians were based on self-reporting; hence, recall bias was inevitable. Second, there is a higher possibility that patients who underwent a less-invasive procedure during hospitalization and have survived long enough after the index event would be able to undergo clinical examinations and interviews, so we do not have complete data from more-severe patients. Therefore, our results might overemphasize the level of GRRFI or PRRFI. However, with the strong threshold of GRRFI ≥ 80% applied in our study, compared with the threshold provided in other study, which is PLA (a physician’s lifestyle advice) ≥50% [17], we believe that the above-mentioned unmeasured confounding factors were controlled in the study. Our study was conducted in the Polish population, and, thus, our results and clinical implications may not be generalizable to other European populations or regions due to different risk profiles, age compositions of disease, and distributions of lifestyle factors. Hence, our study should be interpreted with caution, and further research is required to overcome these limitations. Finally, the multivariable model included only the known CVD risk factors collected in the study; we were unable to control for unmeasured confounding, thus leaving room for residual confounding.

The novelty of the presented study relies on the multiparameter approach to the patient–doctor communication process. It has to be emphasized that there is no successful “one type fits all” strategy, and, each time communication is performed, in order to be effective, it must include a recipient’s characteristics, as well as an approach towards one’s health, health literacy, and trust in healthcare. These factors have been strikingly important in the recent efforts to increase vaccination frequency during the COVID-19 pandemic [32]. Moreover, we point out that the patients with the poorest recollection of information are the oldest (which could be expected), but they are also the patients with fewer comorbidities, who also deserve attention and ways to receive appropriate health information. Elderly patients, often excluded from modern sources of information, such as the Internet, require particular visual aids to improve their awareness of risk factors, as well as the involvement of their closest family members or caretakers.

## 5. Conclusions

This study revealed that the prevalence of a patient’s recollection of risk factor information was higher in patients with multiple comorbidities and that these patients significantly change their lifestyle behaviors. This suggests that physicians are focused on informing patients with multiple comorbidities. A secondary prevention setting should include modern preventive cardiology programs with multidisciplinary teams of healthcare professionals to address all aspects of lifestyle and risk factor management, hence reducing the risk of adverse cardiovascular events. Following this, country-specific coronary heart disease managed-care programs should be implemented based on the needs of particular subpopulations.

### Recommendations

The knowledge from our study provides several clinical implications. Our investigation on the determinants of the recollection of risk factor information in chronic coronary syndrome patients revealed that slightly more than half of the patients, especially those with multiple comorbidities, recollected risk factor information at a high threshold, about ≥80%. This indicates that physicians were more concerned with patients with multiple risk factors. Therefore, it is important to leverage an approach with innovative ways to help convey information and ensure the long-term effectiveness of communication. Our findings have implications for understanding the effectiveness of patient–doctor interactions and patients’ compliance with a physician’s information. Notably, communication skills are crucial to successful medical practice, which greatly impacts patients’ satisfaction, compliance, and outcomes. Importantly, a modern secondary prevention program should be designed and include effective communication strategies, hence encouraging vigorous behavioral changes.

## Figures and Tables

**Figure 1 ijerph-19-06416-f001:**
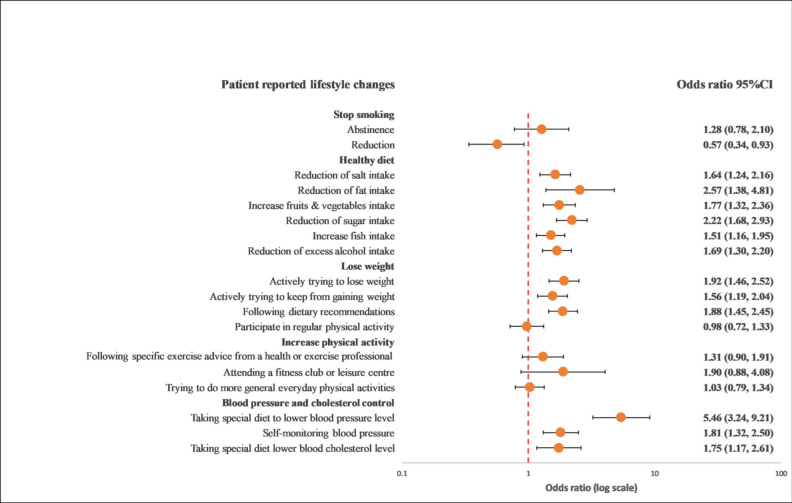
Univariable logistics regression analysis between the recollection of risk factors and patients’ reported lifestyle changes. The model included the recollection of risk factor information (PRRFI = 0 and GRRFI = 1) as the independent variable and patients’ reported lifestyle changes (no = 0 and yes = 1) as the dependent variables. PRRFI: poor recollection of risk factor information; GRRFI: good recollection of risk factor information.

**Figure 2 ijerph-19-06416-f002:**
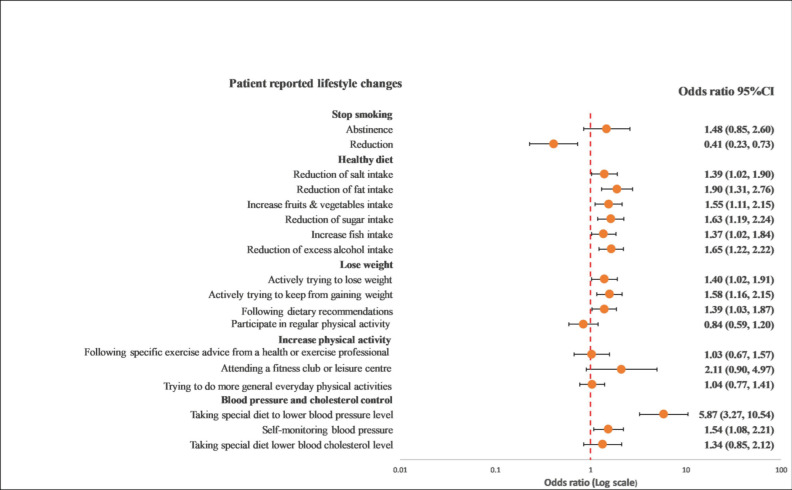
Multivariable logistics regression analysis between the recollection of risk factors and patients’ reported lifestyle changes. The model included the recollection of risk factor information (PRRFI = 0 and GRRFI = 1) as the independent variable, patients’ reported lifestyle changes (no = 0 and yes = 1) as the dependent variables, and age at the index event (continuous), gender, the index event, and obesity, as well as diabetes at hospitalization, as the covariates. PRRFI: poor recollection of risk factor information; GRRFI: good recollection of risk factor information.

**Figure 3 ijerph-19-06416-f003:**
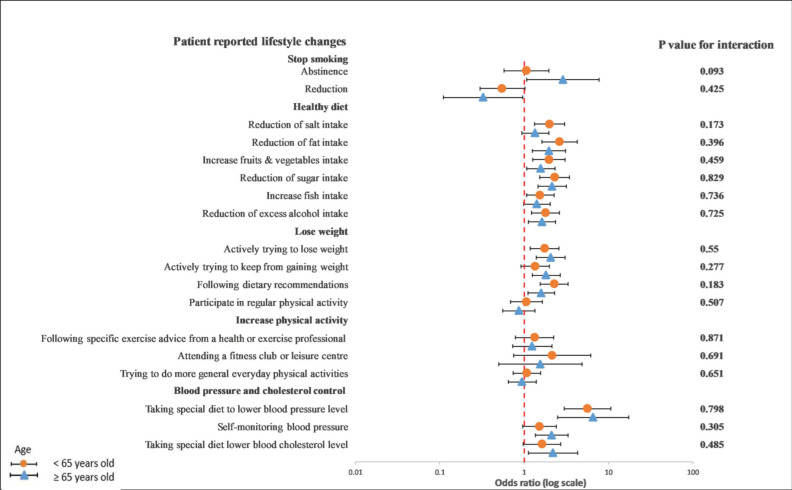
Univariable logistics regression analysis between the recollection of risk factors and patients’ reported lifestyle changes in younger and older patients. The *p*-value corresponds to the interaction between younger and older patients. The model included the recollection of risk factor information (PRRFI = 0 and GRRFI = 1) as the independent variable and patients’ reported lifestyle changes (no = 0 and yes = 1) as the dependent variables. PRRFI: poor recollection of risk factor information; GRRFI: good recollection of risk factor information.

**Figure 4 ijerph-19-06416-f004:**
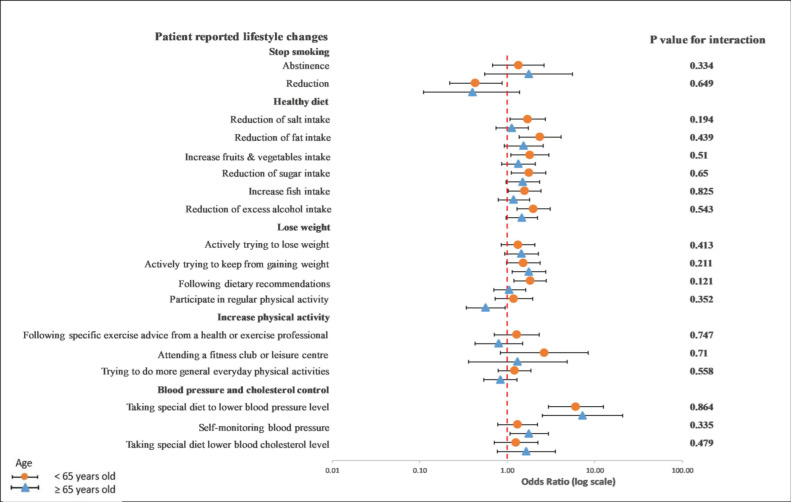
Multivariable logistics regression analysis between the recollection of risk factors and patients’ reported lifestyle changes. The *p*-value corresponds to the interaction between younger and older patients. The model included the recollection of risk factor information (PRRFI = 0 and GRRFI = 1) as the independent variable, patients’ reported lifestyle changes (no = 0 and yes = 1) as the dependent variables, and gender, the index event, and obesity, as well as diabetes at hospitalization, as covariates. PRRFI: poor recollection of risk factor information; GRRFI: good recollection of risk factor information.

**Figure 5 ijerph-19-06416-f005:**
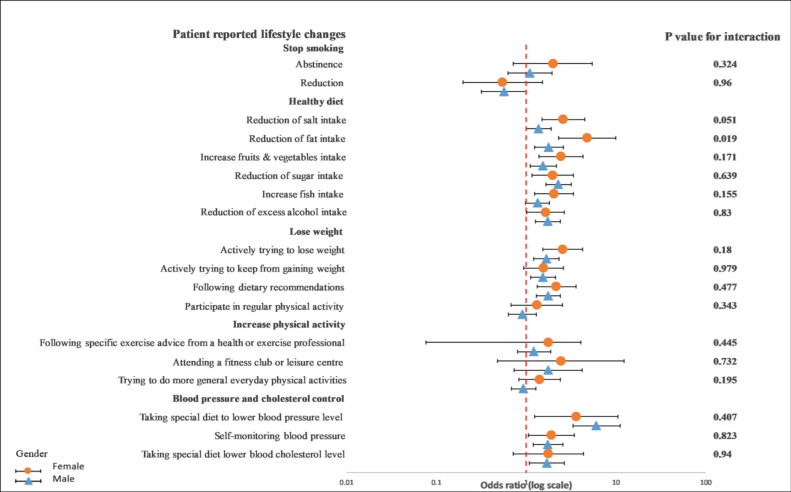
Univariable logistics regression analysis between the recollection of risk factors and patients’ reported lifestyle changes in female and male patients. The *p*-value corresponds to the interaction between female and male patients. The model included the recollection of risk factors (PRRFI = 0 and GRRFI = 1) as the independent variable and patients’ reported lifestyle changes (no = 1 and yes = 1) as the dependent variables. PRRFI: poor recollection of risk factor information; GRRFI: good recollection of risk factor information.

**Figure 6 ijerph-19-06416-f006:**
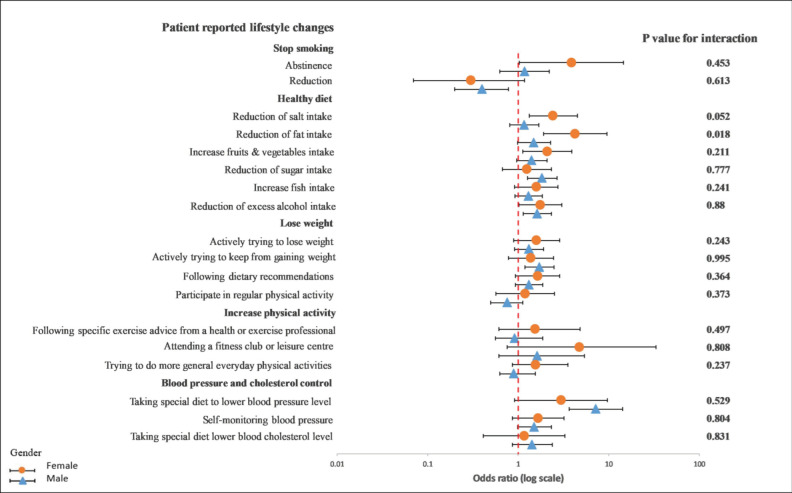
Multivariable logistics regression analysis between the recollection of risk factors and patients’ reported lifestyle changes. The *p*-value corresponds to the interaction between female and male patients. The model included the recollection of risk factors (PRRFI = 0 and GRRFI = 1) as the independent variable, patients’ reported lifestyle changes (no = 0 and yes = 1) as the dependent variables, and age at the index event (continous), the index event, and obesity, as well as diabetes at hospitilization, as the covariates. PRRFI: poor recollection of risk factor information; GRRFI: good recollection of risk factor information.

**Table 1 ijerph-19-06416-t001:** Demographic and Clinical Characteristics of Patients’ Recollection of CVD Risk Factor Information from Physicians in Chronic Coronary Syndrome Patients.

Characteristics	Total Patients *n* = 946	Physician Risk Factor Information Recollection		*p*-Value	Multivariate Odds Ratio (95% CI) ^a^
		Poor Recollection *n* = 445	Good Recollection *n* = 501		
**Age Group, Years**				0.194	
<54	104 (11.0)	51 (11.5)	53 (10.6)		1.00
55–64	349 (36.9)	151 (33.9)	198 (39.5)		0.97 (0.59–1.59)
65–74	379 (40.1)	181 (40.7)	198 (39.5)		0.59 (0.36–0.98)
≥75	114 (12.1)	62 (13.9)	52 (10.4)		0.51 (0.27–0.96)
**Gender**				0.286	
Female	273 (28.9)	121 (27.2)	152 (30.3)		
Male	673 (71.1)	324 (72.8)	349 (69.7)		-
					-
Center				0.756	
Nonteaching hospitals	162 (17.1)	78 (17.5)	84 (16.8)		-
Teaching hospitals	784 (82.9)	367 (82.5)	417 (83.2)		-
**Education Level**				0.304	
Primary education	129 (13.6)	55 (12.4)	74 (14.8)		-
Middle education	640 (67.7)	312 (70.1)	328 (65.5)		-
Higher education	177 (18.7)	78 (17.5)	99 (19.8)		-
**Recruiting Event**				0.087	
CABG	38 (4.0)	21 (4.7)	17 (3.4)		-
PCI	355 (37.5)	157 (35.3)	198 (39.5)		-
ST-EMI	148 (15.6)	73 (16.4)	75 (15.0)		-
Non-ST-EMI	200 (21.1)	84 (18.9)	116 (23.2)		-
UA	205 (21.7)	110 (24.7)	95 (19.0)		-
**Smoking Habit**				0.152	
Nonsmoker	422 (44.6)	191 (42.9)	231 (46.1)		-
Former smoker	296 (31.3)	134 (30.1)	162 (32.3)		-
Current smoker	228 (24.1)	120 (27.0)	108 (21.6)		-
**Obesity**				<0.001	
	49				1.00
Yes	370 (39.1)	87 (19.6)	283 (56.5)		4.41 (3.09–6.30)
Unknown	180 (19.0)	87 (19.6)	93 (18.6)		1.07 (0.72–1.59)
Weight, kg	82.75 (74.0, 94.0)	79.0 (70.52, 88.0)	89.0 (79.0, 98.0)	<0.001	-
BMI, kg/m^2^	29.03 (26.27, 32.16)	27.73 (25.51, 29.74)	31.0 (28.82, 33.46)	<0.001	-
**Hypertension**				<0.001	
Yes	863 (91.2)	389 (87.4)	474 (94.6)		
SBP, mm Hg	136.0 (125.0, 150.0)	130.0 (122.2, 146.75)	140 (128.0, 153.5)	0.010	-
DBP, mm Hg	80.0 (73.0, 88.0)	80.0 (72.0, 85.0)	80.0 (74.0, 90.0)	<0.001	-
**Hyperlipidemia**				<0.001	
Yes	733 (90.6)	328 (87.5)	405 (93.3)		-
LDL-C, mmol/L	2.43 (1.86, 3.39)	2.50 (1.83, 3.39)	2.40 (1.83, 3.34)	0.474	-
HDL-C, mmol/L	1.14 (0.95, 1.39)	1.21 (0.98, 1.45)	1.11 (0.93, 1.34)	0.003	-
Triglycerides, mmol/L	1.30 (0.97, 1.86)	1.22 (0.87, 1.71)	1.38 (1.05, 1.92)	0.007	-
**Diabetes**				<0.001	
No	628 (66.4)	370 (83.1)	258 (51.5)		1.00
Yes	318 (33.6)	75 (16.9)	243 (48.5)		4.16 (2.96–5.84)
**Medication Prescribed**					
Antiplatelets	934 (98.7)	441 (99.1)	493 (98.4)	0.338	-
Beta-blockers	867 (91.6)	406 (91.2)	461 (92.0)	0.665	-
ACEI/sartan	722 (76.3)	358 (80.4)	364 (72.7)	0.005	0.65 (0.45–0.94)
Statins	894 (94.5)	420 (94.4)	474 (94.6)	0.878	-
Calcium channel blockers	260 (27.5)	92 (20.7)	168 (33.5)	<0.001	1.47 (1.04–2.09)
Diuretics	466 (49.3)	188 (42.2)	278 (55.5)	<0.001	1.41 (1.03–1.91)
Anticoagulants	139 (14.7)	57 (12.8)	82 (16.4)	0.124	-

All variables were compared using the chi-square test. ST-EMI: ST-elevation myocardial infarction; CABG: coronary artery bypass graft; PCI: percutaneous coronary interventions; UA: unstable angina; primary educational level denoted the completion of primary and secondary school; middle education level denoted the completion of high school or technical/vocational training; and higher education level denoted the completion of college or postgraduate studies. ^a^ Derived using a backward logistic regression model that included all variables, with *p* < 0.25. The final model comprised age (continuous), gender, center, region, the index event, obesity, hypertension, hyperlipidemia, diabetes, ACEI/sartan, calcium channel blockers, diuretics, and anticoagulants. A missing value was treated as unknown in the multivariable logistic model.

**Table 2 ijerph-19-06416-t002:** Determinants of Recollection of a Physician’s Information on Risk Factors in Chronic Coronary Syndrome Patients.

	Recollection of a Physician’s Information on Risk Factors					
Dependent Variables	Poor Recollection *n* = 445	Good Recollection *n* = 501	OR ^a^	*B* ^b^	95% CI	*p*-Value
Risk Factor Goals						
Stopped smoking ^c^	58 (50.9)	56 (49.1)	1.48	-	0.85 to 2.60	0.161
Reduction in smoking ^c^	77 (61.1)	49 (38.9)	0.41	-	0.23–0.73	0.003
Increased physical activity ^d^	68 (15.3)	63 (12.6)	0.77	-	0.50–1.19	0.252
**BMI, kg/m^2^**						
<25	108 (24.3)	35 (7.0)	0.45		0.28–0.71	0.001
<30	332 (74.6)	217 (43.3)	0.52	*-*	0.37–0.74	0.000
Blood pressure on target ^e^	393 (88.9)	427 (85.4)	0.65	*-*	0.41–1.01	0.058
LDL cholesterol on target ^f^	175 (39.8)	180 (36.1)	0.70	*-*	0.52–0.96	0.028
HbA1c on target ^g^	357 (93.5)	351 (77.8)	0.63	*-*	0.36–1.08	0.097
Antiplatelet, *n*%						
*Aspirin*	401 (90.1)	439 (87.6)	0.75	*-*	0.47–1.21	0.247
*Clopidogrel*	222 (49.9)	233 (46.5)	0.76	*-*	0.56–1.02	0.068
**Lipid-Lowering Drugs, *n*%**						
*Atorvastatin*	292 (65.6)	316 (63.1)	0.90	*-*	0.66–1.23	0.476
*Rosuvastatin*	90 (20.2)	120 (24.0)	1.14	*-*	0.80–1.62	0.456
ACE inhibitors, *n*%	311 (69.9)	354 (70.7)	1.01	*-*	0.73–1.39	0.951
Beta-blockers, *n*%	385 (86.5)	457 (91.2)	1.16	*-*	0.72–1.86	0.528
**Medication Adherence ^h^**						
Lipid-lowering drug > 75% intake	385 (88.1)	455 (92.9)	1.46		0.87–2.42	0.145
Antihypertensive drug > 75% intake	372 (85.7)	463 (94.1)	1.80	-	1.07–3.03	0.026
Glucose-lowering drug > 75% intake	103 (23.7)	242 (49.3)	0.81	-	0.48–1.38	0.458
**Quality of life**						
HADS—anxiety, mean ± SD	5.70 ± 3.77	5.90 ± 3.54	-	0.35	−0.17 to 0.88	0.184
HADS—depression, mean ± SD	5.45 ± 3.66	5.50 ± 3.36	-	−0.01	−0.51 to 0.49	0.964
Heart QoL ^f^ global, mean ± SD	29.06 ± 8.41	27.36 ± 9.02	-	−2.08	−3.75 to −0.42	0.014
Heart QoL emotional, mean ± SD	7.70 ± 2.30	7.39 ± 2.31	-	−0.49	−0.94 to −0.05	0.027
Heart QoL physical, mean ± SD	21.37 ± 7.07	20.0 ± 7.66	-	−1.57	−2.97 to 00.16	0.029
VAS overall	0.86 ± 0.12	0.84 ± 0.14	-	−0.02	−0.05 to 0.00	0.068

Proportions for categorical variables, %; BMI, body mass index; CI, 95 percent confidence interval; OR, odds ratio; β coefficient; and significant *p*-value, < 0.05. ^a^: multivariable logistics regression adjusted for age at the index event (continuous), gender, the index event, obesity, and diabetes. ^b^: multivariable linear regression adjusted for age at the index event (continuous), gender, the index event, obesity, and diabetes. ^c^: only patients that reported smoking the month before the index event were included. ^d^: active physical activity denoted at least 20 min once or twice a week. ^e^: blood pressure ≤ 140/90 mmHg. ^f^: LDL-C < 1.80 mml/L. ^g^: HBA1c < 7%. ^h^: patient reported taking drugs most of time (75%), nearly all of the time (90%), or all the time (100%). Abbreviations: BP, blood pressure; HbA1c, glycated hemoglobin; HADS, Hospital Anxiety and Depression Scale; heart QoL, heart-related quality of life; and VAS, visual analogue score.

**Table 3 ijerph-19-06416-t003:** Changes in Clinical Parameters Compared with the Status of Physician Risk Factor Information Recollection in Chronic Coronary Syndrome Patients.

Dependent Variables	Physician Risk Factor Information Recollection						Difference in Change from Regression (Crude Model)	*p-*Value	Difference in Change from Regression (Adjusted Model)	*p-*Value ^a^
	Poor Recollection (*n* = 445)			Good Recollection (*n* = 501)						
Clinical Parameters	Baseline	Interview	Mean Change in Baseline to Interview	Baseline	Interview	Mean Change in Baseline to Interview				
BMI, kg/m^2^	27.58 ± 3.99	27.73 ± 3.97	0.15 ± 2.23	30.95 ± 4.25	31.12 ± 4.30	0.16 ± 2.23	0.01 (−0.30 to 0.33)	0.923	0.47 (0.11 to 0.83)	0.010
SBP, mm Hg	135.37 ± 19.87	132.29 ± 19.12	−3.08 ± 22.18	139.98 ± 21.75	134.64 ± 18.32	−5.34 ± 24.37	−2.25 (−5.32 to 0.81)	0.150	−2.50 (−5.99 to 0.98)	0.159
DBP, mm Hg	79.44 ± 10.64	79.13 ± 10.28	−0.31± 12.13	81.71 ± 12.31	80.77 ± 10.92	−0.94 ± 14.38	−0.63 (−2.39 to 1.12)	0.478	−0.13 (−2.11 to 1.85)	0.896
LDL-C, mmol	2.69 ± 1.11	2.20 ± 0.91	−0.49 ± 1.12	2.66 ± 1.16	2.24 ± 1.01	−0.42 ± 1.20	0.06 (−0.10 to 0.23)	0.427	−0.05 (−0.24 to 0.13)	0.588
HDL-C, mmol/L	1.26 ± 0.44	1.34 ± 0.38	0.08 ± 0.36	1.16 ± 0.39	1.27 ± 0.36	0.10 ± 0.33	0.02 (−0.02 to 0.07)	0.369	0.01 (−0.04 to 0.07)	0.577
Triglycerides, mmol/L	1.42 ± 0.90	1.38 ± 0.85	−0.03 ± 0.76	1.62 ± 0.96	1.54 ± 0.84	−0.08 ± 0.96	−0.05 (−0.17 to 0.07)	0.418	−0.05 (−0.19 to 0.08)	0.455

Continuous data are presented as means (SD). ^a^: *p*-value derived from the differences between recollection of lifestyle advice and measurement time points. β coefficient for continuous outcomes. Model 1: crude model. Model 2: adjusted for age at the index event (continuous), gender, the index event, diabetes, and obesity.

**Table 4 ijerph-19-06416-t004:** Change in Medication Intake Compared with the Status of a Physician’s Risk Factor Information Recollection in Chronic Coronary Syndrome Patients.

Dependent Variables	Physician Risk Factor Information Recollection						Estimate of Difference in Change From Regression Model 1	*p-*Value	Estimate of Difference in Change from Regression Model 2	*p-*Value ^a^
	Poor Recollection (*n* = 445)			Good Recollection (*n* = 501)						
Medication	Baseline	Interview	Percentage Change in Baseline to Interview	Baseline	Interview	Percentage Change in Baseline to Interview				
Antiplatelets	441 (99.1)	413 (93.7)	−5.4 %	493 (98.4)	463 (93.9)	% 5.6	1.04 (0.61–1.78)	0.867	1.05 (0.57–1.92)	0.87
Beta-blockers	406 (91.2)	369 (90.9)	−0.3%	461 (92.2)	441 (95.7)	1.9%	2.21 (1.26–3.87)	0.006	1.54 (0.81–2.91)	0.182
ACE inhibitors	358 (80.4)	289 (80.7)	0.3%	364 (72.7)	313 (86.0)	2.1%	1.46 (0.98–2.17)	0.058	1.47 (0.94–2.30)	0.091
Statins	420 (94.4)	380 (90.5)	−3.9%	474 (94.6)	434 (91.6)	0.4%	1.14 (0.72–1.80)	0.571	1.07 (0.63–1.80)	0.795

Categorical data are presented as numbers (percentages). ^a^: odds ratio for categorical outcomes. Model 1: crude model. Model 2: adjusted for age at the index event (continuous), gender, the index event, diabetes, and obesity.

## Data Availability

Data available on request only for scientific purposes.

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
