# Peer review of "Recollection of Physician Information about Risk Factor and Lifestyle Changes in Chronic Coronary Syndrome Patients"

_ijerph, 2022, doi:10.3390/ijerph19116416_

Round 1

Reviewer 1 Report

The paper can be published after a revision. Please follow my comments as below.

  1. Please improve this study with a articulation of the novelty and significance.
  2. All abbreviations need to be defined in a separate section.
  3. Please revise the texts throughout the manuscript because it contains too errors.
  4. Please give some limitations and recommendations from this study

Thank you!

Reviewer 2 Report

Authors investigated the prevalence of patient’s recollection of risk factor information was higher in patients with multiple comorbidities and those patients have significantly change their lifestyle behaviors. This study is important how patients can change their lifestyle and compliance to physician information. The reviewer found a lot of mistakes in this manuscript. Please confirm as below.

  1. Please add the approval number when authors applied to use POLASPIRE database.
  2. What was the meaning of SINCE (page 3)? Please clarify it in methods. The meanings of GRRFI and PRRFI (page 4) also should be clarified in methods.
  3. Please show the percentage of participants in Obesity/unknown, Hypertension/All, and Hyperlipidemia/All in Table 1. In addition, it is enough to show only “Yes” group among obesity, hypertension, hyper lipidemia, diabetes in Table 1.
  4. In Table 2;
    1. Did BMI < 30 mean 25 ≤ BMI < 30?
    2. The percentage was incorrect in “Blood pressure on target”, “LDL cholesterol on target”, “HbA1c on target”, “Aspirin”, “Clopidogrel”, ”Atorvastatin”, ”Rosuvastatin”, ”ACE inhibitors”, ”Beta-blockers”, “Lipid lowering drug > 75 % intake”, ”Anti-hypertensive drug >75 % intake”, and ”Glucose lowering drug >75 % intake”.
    3. The number of participants in poor recollection group exceeded the total number. Please confirm all numbers. In addition, please show the column data in changes in smoking habit.
  5. All percentages were incorrect in Table 4.
  6. The reviewer thought that No = 0 was correct in the legend of Figure 1, 2, 3, 4, 5, and 6.
